# *Ophelimus migdanorum* Molina-Mercader sp. nov. (Hymenoptera: Eulophidae): Application of Integrative Taxonomy for Disentangling a Polyphenism Case in *Eucalyptus globulus* Labill Forest in Chile

**Gloria Molina-Mercader** [1,*], **Andrés O. Angulo** [2], **Tania S. Olivares** [2], **Eugenio Sanfuentes** [3], **Miguel Castillo-Salazar** [3], **Eladio Rojas** [4], **Oscar Toro-Núñez** [5], **Hugo A. Benítez** [6] and **Rodrigo Hasbún** [7,*]

[1] *MIP*lagas Ltda., Avda. Las Rosas 1973, Huertos Familiares, San Pedro de La Paz, Concepción 4130000, Chile
[2] Departamento De Zoología, Facultad de Ciencias Naturales y Oceanográficas, Universidad de Concepción, Casilla 160-C, Concepción, Concepción 4030000, Chile
[3] Facultad de Ciencias Forestales, Universidad de Concepción, Casilla 160-C, Concepción, Concepción 4030000, Chile
[4] Laboratorio Regional Servicio Agrícola y Ganadero, Unidad de Entomología., Osorno 5290000, Región de Los Lagos, Chile
[5] Departamento de Botánica, Facultad de Ciencias Naturales y Oceanográficas, Universidad de Concepción, Casilla 160-C, Concepción, Concepción 4030000, Chile
[6] Departamento de Biología, Facultad de Ciencias, Universidad de Tarapacá, Arica 1000000, Región de Arica y Parinacota, Chile
[7] Laboratorio de Epigenética, Departamento de Silvicultura, Facultad de Ciencias Forestales, Universidad de Concepción, Concepción 4030000, Chile
[*] Correspondence: gloria.molina@miplagas.cl (G.M.-M.); rodrigohasbun@udec.cl (R.H.); Fax: + 56-9-56187684 (G.M.-M.)

**Abstract:** In 2003, a new gall-inducing wasp of the genus *Ophelimus* was detected in the Valparaíso Region (Chile), affecting tree plantations of *Eucalyptus globulus* Labill and *Eucalyptus camaldulensis* Dehnh. *Since then Ophelimus* has been frequently detected in different plantations in Chile, covering a widespread area. A preliminary collaborative study suggests that the micro-wasp detected should be classified as a new *Ophelimus* species. In this paper, using an integrative approach (including genetic, morphological, and behavioral data), we addressed the delimitation and description of this new species. This study involved the use of brood adult specimens, raised at the laboratory of *MIP*lagas Ltda., from infested twigs of *E. globulus* collected in several localities between of Valparaíso and Los Lagos Regions (Chile). Morphological structures were described according to current Eulophidae taxonomic keys, as well as additional traits, such as gall morphology and behavior. Genetic characterization was implemented using a phylogenetic approach, based on a 648 bp specific fragment of the mitochondrial Cytochrome Oxidase I gene (*COI 5* region) obtained from collected specimens and available databases (Genbank, NCBI, and BOLDSystem). Specifically, distinctive patterns of variation were detected in traits like gall and antennae morphology, growth habit trends, and a notorious polyphenism in the setae from the sub marginal vein. Overall evidence suggests that this new entity should be considered a new species in *Ophelimus*, which is henceforth named *Ophelimus migdanorum* Molina-Mercader.

**Keywords:** new *Ophelimus* species; Eulophidae; wasp galls; Chile; *Eucalyptus sp.*; polyphenism

## 1. Introduction

Species of the genus *Eucalyptus* L'Hér. (Myrtaceae) are native to Australia, Tasmania, and nearby islands, being planted worldwide due to their common commercial, ornamental, and industrial use. Phytosanitary problems caused by pests and diseases in plantations of *Eucalyptus* sp. outside its natural distribution are of increasing interest because of their impact on the sustainability of forest resources [1–4]. Non-native invasive insects have a significant impact on forestry because they are not accompanied by their principal natural enemies, so can easily reach epidemic levels [5]. Gall inducers of the Eulophidae family—which includes the micro wasp in Hymenoptera, of the genus *Ophelimus*—represent an important radiation from Australia, constituting one of the most important invasive species present in forests [6–9].

The Eulophidae family is one of the largest and most diverse of the super family Chalcidoidea [10]. This family is mainly composed of parasitoid species, which also include predators and phytophage insects, as well as gall inducers responsible for the infection of several forest-dwelling species, including *Eucalyptus* [8,10–12]. This family has more than 4472 species, grouped in 297 genera [8,13] and five subfamilies: Eulophinae, Entedoninae, Tetrastichinae [11], Entiinae, and Opheliminae [10]. Four main features have been described to support the monophyly of Eulophidae: Small, straight, and simple protibial spur; four tarsal segments; marginal vein is usually long; antenna with a maximum of 10 segments, with one to four or rarely five funicular segments [14,15]. In Eulophidae, Opheliminae stands out because the genera *Ophelimus* and *Australsecodes* are the most diversified gall-inducing species of the family [8,10,11].

*Ophelimus Haliday* (1844) includes species of micro wasps (Chalcidoidea: Eulophidae and Opheliminae) which induce galls or abnormal growths in different species of *Eucalyptus* [3,8,12]. These are morphologically characterized for presenting female, claviform antenna with 0–4 funicular rings; an anterior wing with a thickened marginal vein; having a marginal post vein; a mesonotum with deep notches throughout the length; the absence of medial striae; and a scutellum without sub-lateral striae [16]. The systematics and taxonomy of *Ophelimus* are poorly understood, as more than 50 species have been only partially described, and the only existing taxonomic key is contained in an unfinished manuscript prepared by the American entomologist Alexandre Arsene Girault (1884–1941) [6].

The origin of *Ophelimus* is Australian [7]. There was a subsequent dispersal to different regions of the world. *Ophelimus maskelli*, has been reported in Italy [17], Spain [18], France [19], Turkey [20], Portugal [21], Argentina (2013) [22], the United States of America (2014) [23], Israel, South Africa, New Zealand, Vietnam, and Indonesia [24]. *O. eucalypti* has been reported in New Zealand [25], Iran, Morocco, Kenya, Uganda [17,18,26], and Greece [27]. Among the hosts of *O. maskelli* and *O. eucalypti*, the tree species *Eucalyptus globulus* Labill, *E. camaldulensis* Dehnhardt and *E. saligna* Sm. have been described, as well as other species of economic importance [6,28]. Strong attacks and gall production lead to intense defoliation, and therefore to a decrease in growth, development, and vigor of susceptible trees [7]. Thus, the presence of *O. maskelii* is usually related to significant economic losses, where severe examples of infestations have reported total defoliation in adult trees of *Eucalyptus* [29]. For example, in New Zealand, defoliation caused by *O. eucalypti* on *E. globulus* has precluded the growth of commercially viable crops [25,30].

Since the defining morphological characters of Eulophidae are widely shared with other chalcidoids (reductions of the usual chalcidoid characteristics) their origin has been suspected to be potentially convergent [10,11,14]. As a result, issues remain in Eulophid taxonomy, particularly in the delimitation of groups at intra-family levels. Morphological characters are not necessarily representative of evolutionary radiation, because of the effect of reversals and parallel evolution. Recent studies [10,11] have attempted to improve this limitation, demonstrating the relevance of employing integrative taxonomy (morphological and molecular data) to generate significant improvement in classifications of this group. Yet, pending status remains for several understudied genus and species of Eulophidae, including *Ophelimus* [10].

A recurrent issue in the use of diagnostic morphological characteristics in insects is their variability driven by adaptive changes. Among the possible mechanisms involved in this response, polyphenism is one of the most frequent, which is defined by the generation of multiple different phenotypes produced by the same genotype. For example, many insect species exhibit facultative wing growth, which reflects a physiological and evolutionary compromise between dispersal and reproduction, triggered by environmental conditions. Environmental conditions can alter the shape, function, and behavior of organisms on short and long-time scales, and even for generations. Polyphenisms are an important reason for the success of insects; allowing them to divide the stages of their life cycle in order to modify different phenotypes that better adapt them to predictable environmental changes (seasonal morphs), with the purpose of coping with temporally heterogeneous environments (dispersal morphs) and dividing work within social groups (eusocial insects) [31–34].

In Chile, *Eucalyptus* forests occur in the mediterranean region, where a growing number of introduced pest species have been detected. The first reports of *Ophelimus* in Chile are from 2003, associated with the production of galls on *E. globulus* and *E. camaldulensis* [35] in the Valparaíso Region, exhibiting a continuous increase in its range of distribution to the Biobío Region by the end of 2009 (author's personal observation). Even though *O. maskelii* seemed to be the most likely cause of galls, preliminary prospection conducted by Molina-Mercader and collaborators, determined that Chilean specimens are substantially different to the description of this taxon (data non-published). Some differentiating characters of the Chilean specimens are two setae in the submarginal vein (SSV), a larger body size, and a light metallic green color. Also, Chilean specimens presented a unique variation of one, two, and three setae from the submarginal vein. The sum of this evidence, plus notorious differences from available descriptions of *O. maskelii* [6], suggests the presence of this and other species of genus *Ophelimus* in Chile.

In that context, the present study had the objective of conducting a morphological, molecular, and behavioral analysis of the specimens collected in Chile, and eliciting their identity within the family Eulophidae and the genus *Ophelimus*. As a result, we expect that this report will help to clarify the species status of Chilean *Ophelimus*, providing a diagnostic description of this taxa and discussing the causes of their phenotypic variations (polyphenism) that make diagnosis a difficult task for the control of this pest against *Eucalyptus*.

## 2. Materials and Methods

### 2.1. Field Sampling

Between August and September 2017, samples were collected from six *E. globulus* plantations showing evidence of *Ophelimus* attack. The plantations are located between the Valparaíso Region in the north, and the Los Lagos Region in the southern part of Chile (Table 1). The sample consisted of three twigs of approximately 50 cm in length, containing galls in the laminae, central ribs, petioles, and stems. The samples were placed in polystyrene bags of 35 cm × 40 cm with absorbent paper, labeled, and sent to *MIP*lagas Ltda. laboratory, where they were processed and analyzed. Each sampling point was georeferenced.

**Table 1.** Location of *Eucalyptus globulus* plantations where *Ophelimus* nov. Sp. samples were collected.

| Location | Region | Coordinates S | Coordinates O |
|---|---|---|---|
| Casa Blanca | Valparaíso | 33°29′3.2″ | 71°31′47.8″ |
| Litueche | O'Higgins | 34°04′33.8″ | 71°48′12.2″ |
| Cauquenes | Maule | 36°04′27.9″ | 72°00′31.0″ |
| Chillán | Biobío | 36°39′21.4″ | 72°23′27.2″ |
| Nueva Imperial | La Araucanía | 38°43′18.8″ | 72°54′47.3″ |
| Fresia | Los Lagos | 41°15′36.0″ | 73°31′51.0″ |

### 2.2. Laboratory Emergence Stimulation

The collected twigs were placed individually in emergence chambers consisting of transparent plastic boxes (20 × 30 × 40 cm) with a lid; on the bottom of the camera, two sheets of absorbent paper were installed. Each breeding chamber was sealed with plastic tape, labeled to maintain traceability, and kept in the laboratory until adult emergence, at a temperature varying between 18 and 22 °C, with a relative humidity of approximately 60% and a photoperiod of 16 hours of light and 8 hours of darkness. The emergence chambers were checked every day and the absorbent paper changed every other day to avoid contamination.

The specimens of *Ophelimus* obtained from the emergence chambers were fixed individually in both 70% ethanol and isopropyl alcohol. Then they were placed in cryopreservation tubes. The first one was used for morphological description and assembly of the type material, while those fixed in isopropyl alcohol were used for molecular characterization.

### 2.3. Morphological Description

The description of the new species is based on adult females and males that emerged from the breeding chamber. Terminology of the morphological structures followed Protasov et al. [6], Burks et al. [10], and Gibson et al. [36] As a measurement and observation instrument, a trinocular flat chromatic stereo mic magnifying glass, (BEL model, Solaris-T-Led) and a Microscope (OPTIKAL B-1000PH) were used (OPTIKA, Microscopes, Via Rigia, 32, 24010, Ponteranica, BG, Italy). For the photographs and measurements, the program OPTIKALS (view version 3.9.0.602) was used.

The specimens for microphotographs were taken to the CMA Biobio Advanced Microscopy Center of Universidad de Concepción, where the Scanning Electron Microscope (SEM) Tescan Vega 3 SBU Easy Probe was used. The equipment was emptied in "high vacuum" mode, using a secondary electron detector. Since the samples were insects and contained a chitin exoskeleton, this facilitated the emission of electrons on the sample. As the samples were unmetallized, they were placed directly in a sample holder covered with a carbon sheet.

A comparative table was prepared for a comparison of the morphological characteristics presented by the individuals from the samples collected in the field, and those described for *O. maskelli* by Burks et al., [10] (Table S1).

### 2.4. Molecular Protocols and Sequence Editing

Genomic DNA was extracted using the DNeasy Plant Mini Kit (Qiagen). PCR amplification, and sequencing of the COI barcode region was performed following standard protocols [37]. PCR and sequencing used a single pair of primers:

LepF1 (ATTCAACCAATCATAAAGATATTGG) and

LepR1 (TAAACTTCTGGATGTCCAAAA AATCA) [29],

which recover a 658 bp region near the 5´ ends of *COI*, including the 648 bp barcode region for the animal kingdom [38]. Sequence editing and alignment were automatically done by mapping Sanger sequencing reads to a reference with Unipro UGENE [39] and manual corrections. DNA sequences have been submitted to GenBank (see Table S5 for accession numbers) and BoldSystem (BIN: ADP0823). DNA voucher specimens were deposited at the Museo de Zoología, Universidad de Concepción, Chile.

Additional DNA sequences from generous *Ophelimus* and outgroup members were downloaded from GenBank, [40] and BoldSystem. This sampling was complemented with the inclusion of *COI* sequences belonging to the Entiineae subtribes: *Astichus, Bellerus, Beornia* and *Euderus* [10]. *Hubbardiella* did not have *COI* sequences at the time of this sampling. Only sequences from species of well-known distribution and identification have been uploaded. Finally, a data matrix with 97 entries (Table S5) was built for subsequent analyzes.

*2.5. Identity of Obtained COI Sequences*

The identity of obtained *COI* sequences was determined using a phylogenetic approach, following the criteria of monophyly employed in the phylogenetic species concept. We preferred this approach, given that generic and specific limits of all studied taxa of *Ophelimus* remained uncertain within Entiineae [10]; hence requiring comparable evidence of common ancestry in case of misidentification across closely related taxa. To achieve this, all sequences were aligned with MAFFT v1.3.7 [41]. Geneious R11 (www.geneious.com) was used to subsequently analyze the resulting dataset with both maximum likelihood (ML) and Bayesian inference (BI) criteria.

The ML analysis was conducted with the program iqtree v1.6.8 [42], for which a TPM3u + F + I + G4 nucleotide substitution model was estimated with a Bayesian information criterion (BIC) test in the inbuilt program ModelFinder [43]. This analysis was conducted using default settings and search parameters. Branch support values were calculated with a non-parametric bootstrap (BS) of 1000 pseudo-replicates. All branches with BS values over 70% (BS > 70) were considered well supported.

BI analysis was conducted with Mr. Bayes v.3.2.6 [44]. Given that the Bayesian approaches can integrate accounted uncertainty in phylogenetic trees and nucleotide substitution models simultaneously [45], a reversible jump MCMC (rjMCMC) search approach= was employed with two independent runs of 1,000,000 iterations, each of 4 chains (3 cold and 1 hot) and using default priors. All resulting trees were summarized in a consensus tree, after discarding 20% of trees as burn-in and retaining all compatible clades. All clades with a posterior probability over 95% were considered as well supported (*pp* > 0.95). Levels of mixing and convergence on splits were scrutinized with Tracer v1.7 [46] and the R package *rwty* [47].

For the estimation of species delimitation with *COI* sequences, two single locus-based approaches were employed. First, both ML and BI summary trees were used to delimit entities based on a multi-rate Poisson tree process (mPTP, [48]). This method is a technical improvement compared to the Poisson tree process (PTP, [49]), which involves the modeling of the branching process on the number of accumulated expected substitutions between speciation events but assumes different rates of speciation events among lineages [48].

Alternatively, an analysis of rates of speciation and neutral coalescence thresholds via the Generalized Mixture Yule Coalescent (GMYC) algorithm was employed for delimitation [50]. As this approach requires an ultrametric tree, a Bayesian tree search and sampling were conducted with a subset of the *COI*s of *Ophelimus* sequences with BEAST 10.0.0 [51]. Following a molecular substitution model predefined with ModelFinder (GTR + G + I), ann MCMC search was conducted on two runs with 10 million iterations, discarding a burn-in of 10%. In this case, a strict molecular clock and a coalescent model of fixed population size were assumed. These models were used under the assumption that a constant process of generation in populations is produced among targeted lineages, which are currently in a continuous process of speciation and differentiation. Mixing and convergence of splits were equally controlled using the same programs and protocols employed for BI phylogenetic analyses. All resulting trees were summarized in a maximum clade credibility (MCC) tree on median clade heights, which was calculated with TreeAnotator v1.10.0 (available in http://beast.community/treeannotator).

For the estimation of species limits, a Bayesian-based approach of GMYC was implemented (bGMYC), which ponders and identifies possible effects of phylogenetic uncertainty in the placement of thresholds along branch lengths for species delimitation [52]. This analysis was conducted over the last 100 MCMC ultrametric trees obtained from each BEAST run (50 trees per-run) and pondered over the previously obtained MCC tree. In that case, the analysis was run with one million iterations and a discarded burn-in of 5000 iterations. Since the choice of a molecular clock can also alter the inference of branch lengths among delimited groups, we also performed a bGMYC analysis with a relaxed lognormal clock for comparative purposes. In this case, both the inference of MCC trees and bGMYC were conducted following the same parameters and settings than those with a strict clock analysis.

*2.6. Behavioral Data*

The size and form of galls induced by Ophelimus specimens in Chile was investigated by collecting leaves with fully developed galls on E. globulus trees distributed throughout the country. Leaves with mature galls were collected between November and January. Also, to characterize the morphological characteristics of adults which emerged from different parts of trees, galls present in laminae, central ribs, petioles, and stems were isolated. In the laboratory, 100 galls of each part were taken from different geographic sectors of Chile.

## 3. Results

In this study, a total of 4632 *Ophelimus* specimens were used, which correspond exclusively to the new *Ophelimus* species detected in Chile, and 58.9% were females. The emergence of these specimens occurred via each of the parenting chambers that represented each of the seven regions sampled.

*3.1. Morphology*

New *Ophelimus* Species Diagnosis

*Ophelimus migdanorum* Molina-Mercader nov. sp. (Figure 1a–c). Diagnosis

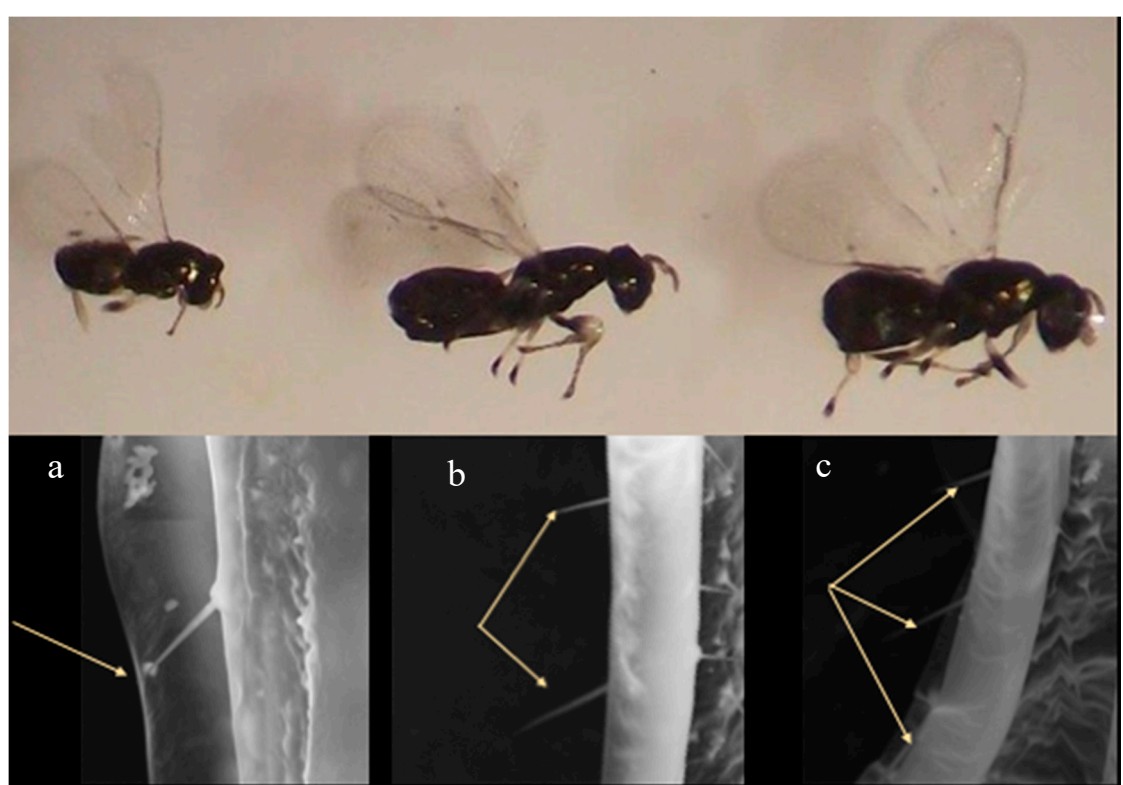

**Figure 1.** *Ophelimus migdanorum* nov sp., with (**a**) one setae on a sub marginal vein, (**b**) two setae on a sub marginal vein, and (**c**) 3 setae on a submarginal vein.

Adult female measures between 0.7 and 1.4 mm (Figure 1a–c), from head to metasoma. Head and thorax are light metallic green; abdomen brown; antennae light brown, dark brown; light chestnut femur; light brown tars; dark brown colored nail; and hyaline wings, with submarginal, marginal, and stigmal veins of a light brown color (Figure 1a–c). Submarginal vein with 1, 2, or 3 satae of uniform size in accordance with the size of every specimen (Figure 1a–c; Table 2).

**Table 2.** Emergence of adults of *Ophelimus migdanorum*, obtained from the areas: Midrib, petiole, leaf blade, and stem. Comparison shown with *Ophelimus maskelli*, according to Branco and Protasov's decription. Oph 1 Seta: *Ophelimus migdanorum* with one seta, and the same for two and three setae, respectively. √: Checkmark shows the places in which they emerged.

|  | *Ophelimus maskelli* | *Ophelimus migdanorum* One Setae | *Ophelimus migdanorum* Two Setae | *Ophelimus migdanorum* Three Setae |
|---|---|---|---|---|
| **Central Rib** |  |  | √ | √ |
| **Petiole** |  | √ | √ | √ |
| **Leaf Blade** | √ | √ | √ | √ |
| **Stems** |  |  | √ | √ |

The antenna presents the scape with cellular crosslinking, elongated on the longitudinal axis. They have a smooth pedicel, 1.5x shorter than the scape; a funicle with three anelli; and two joints that increase in width towards the club nail (Figure 2a). The club is $2 \pm 0.5$ times longer than the funicle; it is globose and has three parts; the last (third), is curved very little, and in the center, it has a long seta (terminal spine) (Figure 2b). Around the second and protruding part of the third segment, four to six longitudinal sensilla are observable (Figure 2b and Figure S2; Tables S2 and S3).

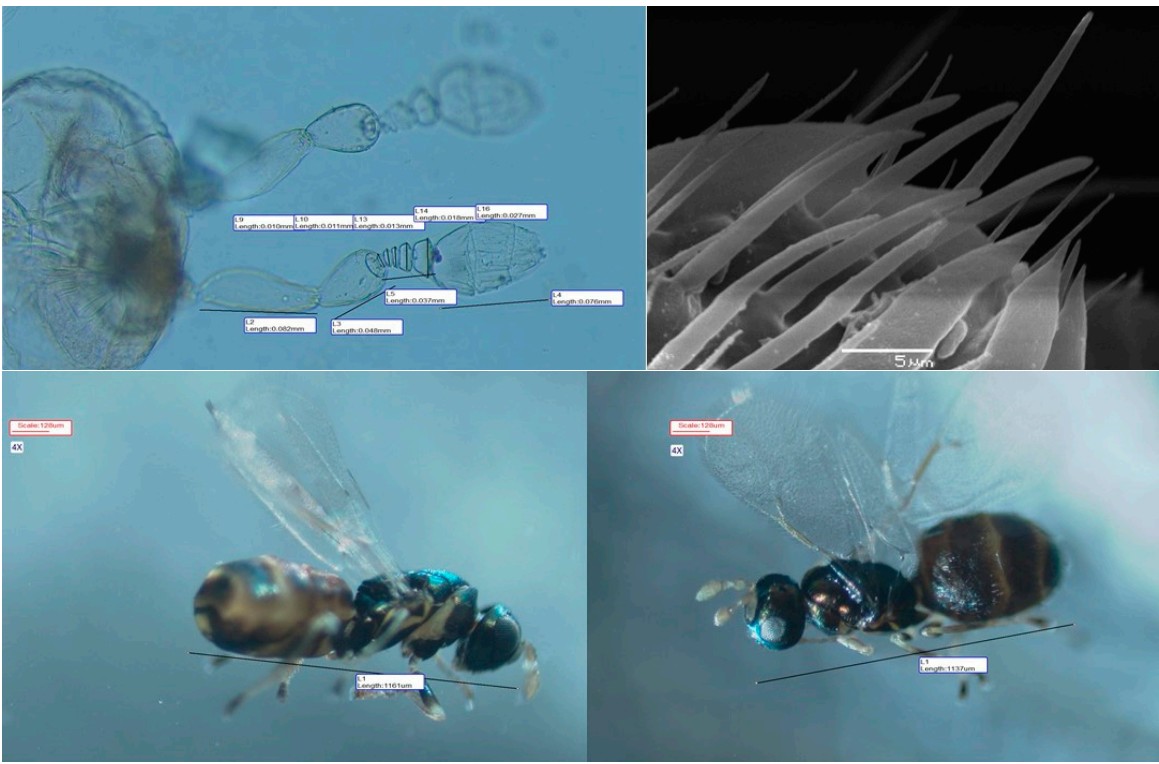

**Figure 2.** *Ophelimus migdanorum nov sp.* key morphological characteristics: (**a**) Antenna and (**b**) top club. (**c**) Female, and (**d**) male.

Male: No difference with respect to females; they can only be differentiated under a stereoscopic magnifying glass and microscopy. The most representative and visual characteristic under these instruments is the curvature present in the thorax and the position of the head, with the head of the male in a straight line with the thorax (Figure 2c,d).

Distribution: Chile, from Valparaíso to Los Lagos regions.

Hosts: *Eucalyptus globulus*.

Etymology: The name designated to this species corresponds to the abbreviation of the first names of the children of the author, Miguel and Daniel.

Provision of the holotype, allotype, and paratypes are in Table S4. (Other measures of this wasp in Tables S2 and S3.)

Figure 1 shows the presence of one, two or three SSV in different specimens of *O. migdanorum*. It can be clearly observed that adults emerged with a number of certain setae. Since no vestigial setae were observed, they did not develop from post gall-emergence. The correlation between the final size of adults emerging from galls and the number of SSVs is high for the individuals belonging to each group (Tables S2 and S3). This may be because the greater the number of SSVs, the greater the size of the individual (Table 3 and Table S3). This phenomenon may be due to a possible case of polyphenism. In order to understand a little more about that phenomenon, galls present in laminae, central ribs, petioles, and stems were isolated, corroborating in 100% of the cases, that *O. migdanorum* nov. sp adults emerged in them. This represents a clear behavioral difference from species *O. maskelli*, in whose case galls are only present in the leaf blade (Table 2). Specifically, in the case of specimens of *O. migdanorum* nov. sp, was observed that adults with one SSV did emerge from central ribs or stems. Therefore, it is easy to confuse them with the behavior of *O. maskelli*, which induces gall formation in the leaf [21,53] and *O. eucalypti* in the petiole and leaf blade [54]. However, Sánchez [18] points out for Spain, that the galls of the petioles or the central rib, correspond to agallicolous species other than *O. eucalypti* [18].

Initially, the specimens collected in Chile did not differ greatly from the description of *O. maskelli*, according to the descriptions given by [6] and [10]. Nonetheless, subsequent observation suggests important differences in the number of SSVs (key 28 of [10]), color, number of rings and funicular segments in the flagellum, and the morphology and location of the gall (Tables 3 and 4). From all the mentioned characteristics, initial descriptions suggested the presence of one SSV for *O. maskelli* and two SSVs for Chilean specimens, yet additional observations from growth chamber specimens suggest a high variability in that characteristic. In this study, specimens with a unique SSV were found, and they could be easily confused with *O. maskelli*, given its notorious variability regarding its having one, two, or three SSVs (Table 2). This observation is confusing based on observations made with French specimens, which revealed that the assignation of the number of SSVs (with individuals having 2, 3, or 4 SSVs) should not be associated to *O. maskelli* [55]. Given that only the SVS number is part of one of the most current keys for the Eulophidae family [10], it is proposed to include several rings and funiculae, as suggested by Protasov et al. [6] for a more complete delimitation of *O. maskelli*, abandoning the use of SSVs for Chilean specimens (Table 4).

In a dendrogram made from the morphological data described in Table 3 (Figure S1), it can be observed that *Ophelimus* morphologically differed less from the *Belerus* than from the *Euderus* or *Astichus* genera. Being between those two, only the characteristics 3 and 30 were different.

## 3.2. Galls

The galls of *Ophelimus migdanorum nov sp.* Chile, *O. maskelli* and *O. eucalypti* are completely different. In the case of *O. maskelli*, they are located only on the sheet, forming a perfect circle (Figure 3a) and in the case of *O. eucalypti*, the galls are on the petiole and the blade, forming a flat gall in the beam of the leaf, and a mound on the underside (Figure 3b) [8]. In contrast, *Ophelimus migdanorum nov sp. Chile*, forms galls on the laminae; the central veins; the petioles; the twigs; the stems of the trees, which are amorphous; and for the adjacent position, where some are elongated (Figure 3c).

**Table 3.** Comparison of *Ophelimus migdanorum* with *Ophelimus maskelli* based on Protasov et al. [6] and La Salle [8].

| Character | Zoom | O. maskelli (um) | O. migdanorum (um) | | | | | | | | | | |
|---|---|---|---|---|---|---|---|---|---|---|---|---|---|
| | | | 1 Setae | | | | 2 Setae | | | | 3 Setae | | | |
| | | | Female (n) | | Male (n) | | Female (n) | | Male (n) | | Female (n) | | Male (n) | |
| Long of the Adult | 4X | 1.026.1 (*) | 865.5 | (± 26.0) | 923.0 | (± 27.9) | 1.0599 | (± 30.6) | 1.1309 | (±33.1) | 1.1569 | (±28.9) | 1.2217 | (±46.7) |
| Width of the Adult | 4X | 234.1 (*) | 249.8 | (± 06.2) | 250 | (± 06.2) | 276.3 | (± 06.8) | 285.9 | (±07.6) | 295.9 | (±05.2) | 319.8 | (±08.0) |
| Long of the Wing | 10X | 900.4 (*) | 768.5 | (± 19.2) | 788 | (± 10.6) | 875 | (± 17.0) | 900.5 | (±25.7) | 934.0 | (±26.6) | 1.0750 | (±49.5) |
| Width of the Wing | 10X | 306.4 (*) | 347.1 | (± 11.4) | 348 | (± 10.8) | 297.3 | (± 10.5) | 413.2 | (±12.6) | 422.1 | (±19.4) | 494.3 | (±25.2) |
| Submarginal | 10X | 304.0 (*) | 213.5 | (± 06.4) | 226 | (± 04.4) | 241.2 | (± 12.7) | 245.3 | (±09.0) | 245.6 | (±13.3) | 300.1 | (±14.7) |
| Marginal | 10X | 153.2 (*) | 157.0 | (± 10.1) | 176 | (± 07.5) | 190.6 | (± 11.5) | 207.7 | (±12.7) | 220.3 | (±21.8) | 255.7 | (±21.2) |
| Postmarginal | 10X | 210.1 (*) | 209.7 | (± 06.1) | 221 | (± 05.4) | 261 | (± 09.5) | 274.2 | (±11.4) | 283.4 | (±13.3) | 315.0 | (±16.6) |
| Stigma | 10X | 99.4 (*) | 64.5 | (± 02.2) | 64.6 | (± 01.7) | 73.8 | (± 01.9) | 74.7 | (±02.1) | 74.6 | (±01.9) | 90.9 | (±04.1) |
| Uncus | 10X | 0.0214 (*) | 18.7 | (± 00.7) | 19.6 | (± 00.7) | 32.9 | (± 11.9) | 22.4 | (±00.9) | 22.0 | (±00.8) | 24.5 | (±01.6) |
| Width/Long of the Adult | | 4.4x (*) | 3.5 | x | 3,7 | x | 3.8 | x | 4.0 | x | 3.9 | x | 3.8 | x |
| Width/Long of the Wing | | 2.9x (*) | 2.2 | x | 2,3 | x | 2.9 | x | 2.2 | x | 2.2 | x | 2.2 | x |
| Submarginal/Marginal | | 2.0x (*) | 1.4 | x | 1.3 | x | 1.3 | x | 1.2 | x | 1.1 | x | 1.2 | x |
| Marginal/Postmarginal | | 0.7x (*) | 0.7 | x | 0.8 | x | 0.7 | x | 0.8 | x | 0.8 | x | 0.8 | x |
| Abdominal Long | 10X | 512.7 (*) | 409.5 | (±14.8) | 430.1 | (±11.7) | 512.0 | (±15.2) | 547.1 | (±17.1) | 568.7 | (±21.2) | 627.1 | (±19.5) |
| Color | 4X | Clear metal green | Dark metalic green | | | | | | | | | | | |
| Number of Setaes in SV | 10X | 1 | 1 a 3 | | | | | | | | | | | |
| Break in Wing Venation | 4X | Yes | No | | | | | | | | | | | |
| Antennae Color | 10X | Dark yellow | light brown | | | | | | | | | | | |
| Number of Rings in the Funicle | 10X | 4 | 3 | | | | | | | | | | | |
| Number of Artejos in Funicle | 10X | 1 | 2 | | | | | | | | | | | |
| Galls | 4X | Leaf blade | Leaf blade, midrib, secondary rib, petiole, twigs, flower cones | | | | | | | | | | | |

(*) measurement Superscript in paper Protasov et al., [6]; (n) indicates measurements made on 30 individuals. Values in parentheses indicate standard error.

**Table 4.** Result of the morphological characterization of *Ophelimus migdanorum* nov sp. in Chile, according to Burks [10] and comparison to other members of Ophelimini and Entiinae. Keys 32 and 33 are additional and highly discriminatory characteristics according to Protosov [6]

| | 1 | 2 | 3 | 4 | 5 | 6 | 7 | 8 | 9 | 10 | 11 | 12 | 13 | 14 | 15 | 16 | 17 | 18 | 19 | 20 | 21 | 22 | 23 | 24 | 25 | 26 | 27 | 28 | 29 | 30 | 31 | 32 | 33 |
|---|---|---|---|---|---|---|---|---|---|---|---|---|---|---|---|---|---|---|---|---|---|---|---|---|---|---|---|---|---|---|---|---|---|
| *Ophelimus maskelli* | 8 | 3 | 0 | 0 | 0 | 0 | 0 | 0 | 0 | 1 | 0 | 0 | 0 | 0 | 1 | 0 | 2 | 0 | 0 | 0 | 0 | 0 | 0 | 0 | 1 | 4 | 0 | 1 | 0 | 1 | 2 | 4 | 1 |
| *Ophelimus nov. sp. 1 SVS* | 8 | 3 | 0 | 0 | 0 | 0 | 0 | 0 | 0 | 1 | 0 | 0 | 0 | 0 | 1 | 0 | 2 | 0 | 0 | 0 | 0 | 0 | 0 | 0 | 1 | 4 | 0 | 1 | 0 | 1 | 2 | 3 | 2 |
| *Ophelimus nov. sp. 2 SVS* | 8 | 3 | 0 | 0 | 0 | 0 | 0 | 0 | 0 | 1 | 0 | 0 | 0 | 0 | 1 | 0 | 2 | 0 | 0 | 0 | 0 | 0 | 0 | 0 | 1 | 4 | 0 | 2 | 0 | 1 | 2 | 3 | 2 |
| *Ophelimus nov. sp. 3 SVS* | 8 | 3 | 0 | 0 | 0 | 0 | 0 | 0 | 0 | 1 | 0 | 0 | 0 | 0 | 1 | 0 | 2 | 0 | 0 | 0 | 0 | 0 | 0 | 0 | 1 | 4 | 0 | 3 | 0 | 1 | 2 | 3 | 2 |
| *Entiinae* | 1 | 2 | 3 | 4 | 5 | 6 | 7 | 8 | 9 | 10 | 11 | 12 | 13 | 14 | 15 | 16 | 17 | 18 | 19 | 20 | 21 | 22 | 23 | 24 | 25 | 26 | 27 | 28 | 29 | 30 | 31 | | |
| *Astichus n. sp* | 8 | 3 | 2 | 0 | 0 | 0 | 0 | 2 | 0 | 1 | 0 | 0 | 0 | 0 | 2 | 0 | 2 | 0 | 0 | 0 | 0 | 1 | 0 | 0 | 1 | 4 | 0 | 3 | 0 | 0 | 2 | | |
| *A.mirissimus* | 8 | 3 | 2 | 0 | 0 | 0 | 0 | 2 | 0 | 1 | 0 | 0 | 0 | 0 | 3 | 0 | 2 | 0 | 0 | 0 | 0 | 1 | 0 | 0 | 1 | 4 | 0 | 3 | 0 | 0 | 2 | | |
| *Bellerus sp.* | 8 | 3 | 2 | 0 | 0 | 0 | 0 | 0 | 0 | 1 | 0 | 0 | 0 | 0 | 1 | 0 | 2 | 0 | 0 | 0 | 0 | 0 | 0 | 0 | 1 | 4 | 0 | 3 | 0 | 0 | 2 | | |
| *Euderus sp.* | 8 | 3 | 2 | 0 | 0 | 0 | 0 | 2 | 0 | 0 | 0 | 0 | 0 | 0 | 3 | 0 | 2 | 0 | 0 | 0 | 0 | 0 | 0 | 0 | 1 | 4 | 0 | 3 | 0 | 0 | 2 | | |

* 32 rings in the funicle; ** 33 number of artejos in funicle.

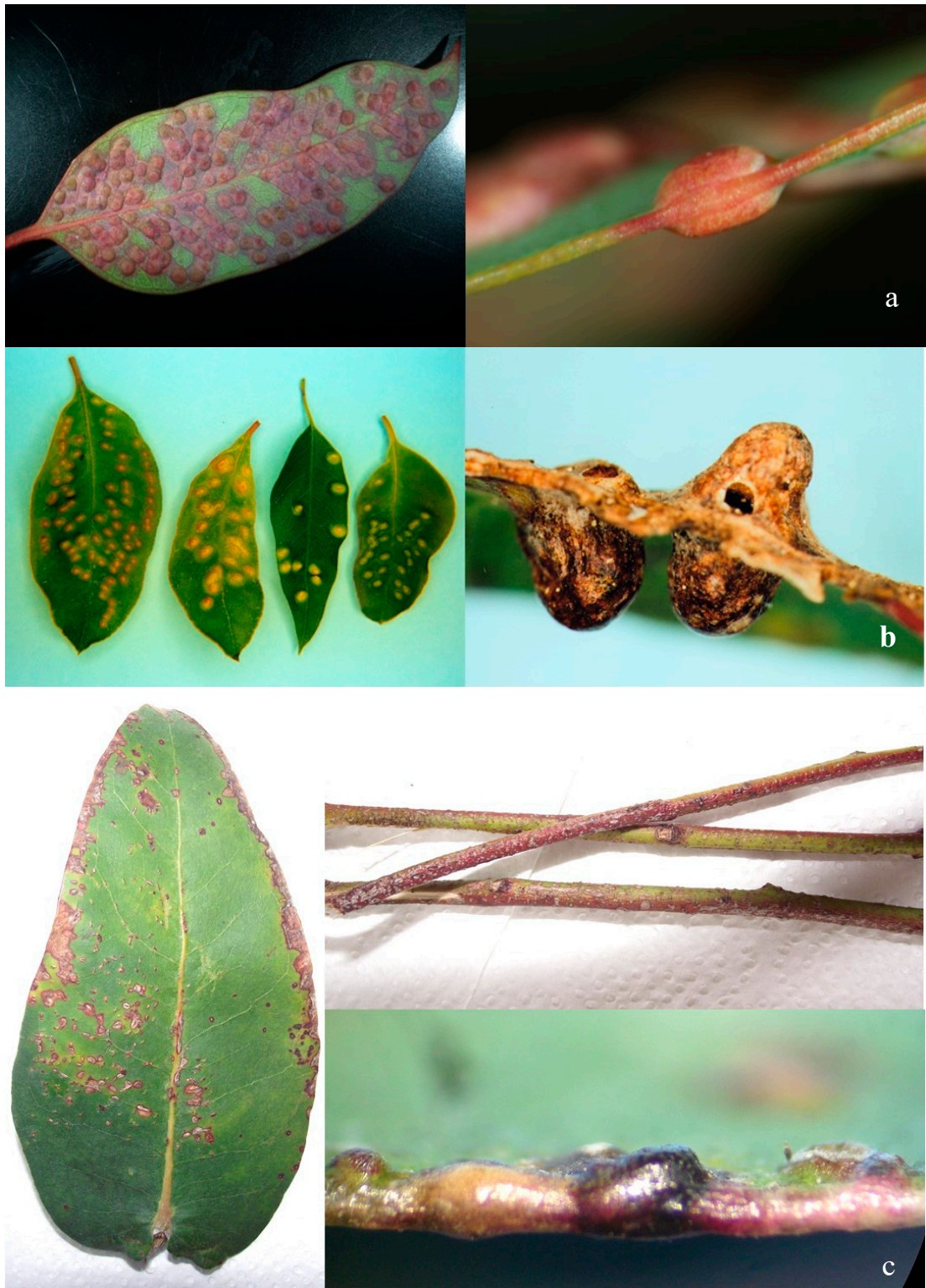

**Figure 3.** Galls formed on *Eucalyptus* spp. (**a**) *Ophelimus maskelli* galls on *Eucalyptus camaldulensis* leaves (Photo Garden, gentleness Dra. Paula Borrajo, Huelva, Spain); (**b**) *Ophelimus eucalypti* galls on *Eucalyptus saligna* leaves (Photo gentleness Dr. John La Salle, in memoriam); (**c**) *Ophelimus nov* sp. Chile, on *Eucalyptus globulus*, leaves and branches.

### 3.3. Sequence Identity COI

The analyses performed with ML and BI revealed that *Ophelimus COI* sequences could not be recovered in a monophyletic group (Figures 4 and 5). In both cases, the sequences of *Ophelimus migdanorum nov sp.,* Chile appear separated from the rest of the *Ophelimus* sequences, being significantly grouped with the sequences of Australian origin (Figures 4 and 5). The rest of the *Ophelimus* sequences, and those registered as *O. maskelli* and *Ophelimus mediterraneus*, are grouped monophilically, only through significant supporting analyses conducted with BI. According to other previously published results [10], the monophyly of the sequences belonging to the Eutiineae subtribe and the basal position of Astichus are confirmed (Figures 4 and 5). The analyses carried out with both criteria were mostly concordant in topology and support, detecting significant differences only in the length of branches and in the support of certain individual groups (e.g., Clade *Ophelimus mediterraneus* and *Ophelimus maskelli*).

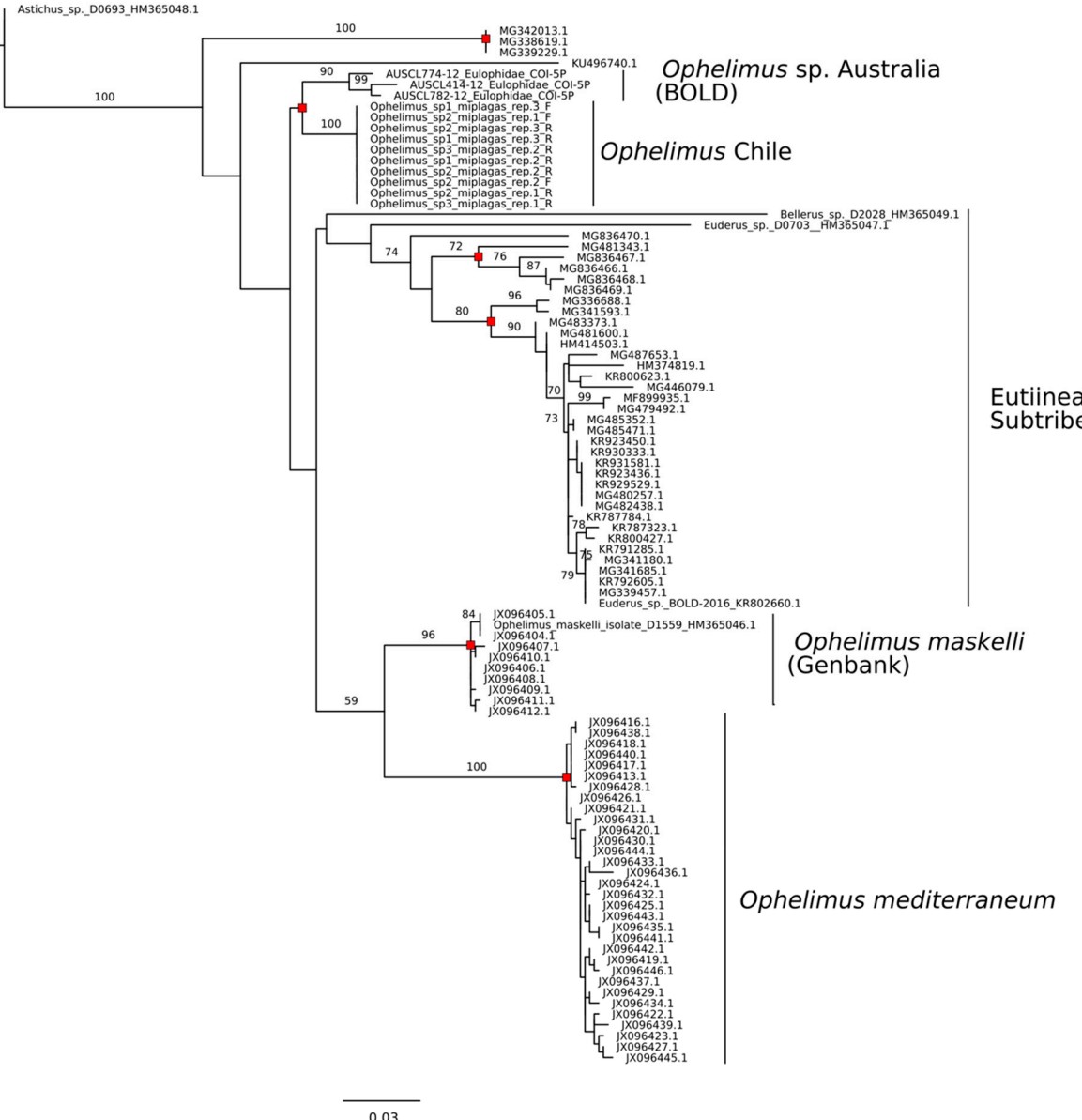

**Figure 4.** Maximum likelihood (ML) tree (log-likehood = −3745.5244) from iqtree. Red squares represent proposed groups made by a multi-rate Poisson tree process (mPTP). Significant bootstrap values (*BP* > 70) are shown above branches.

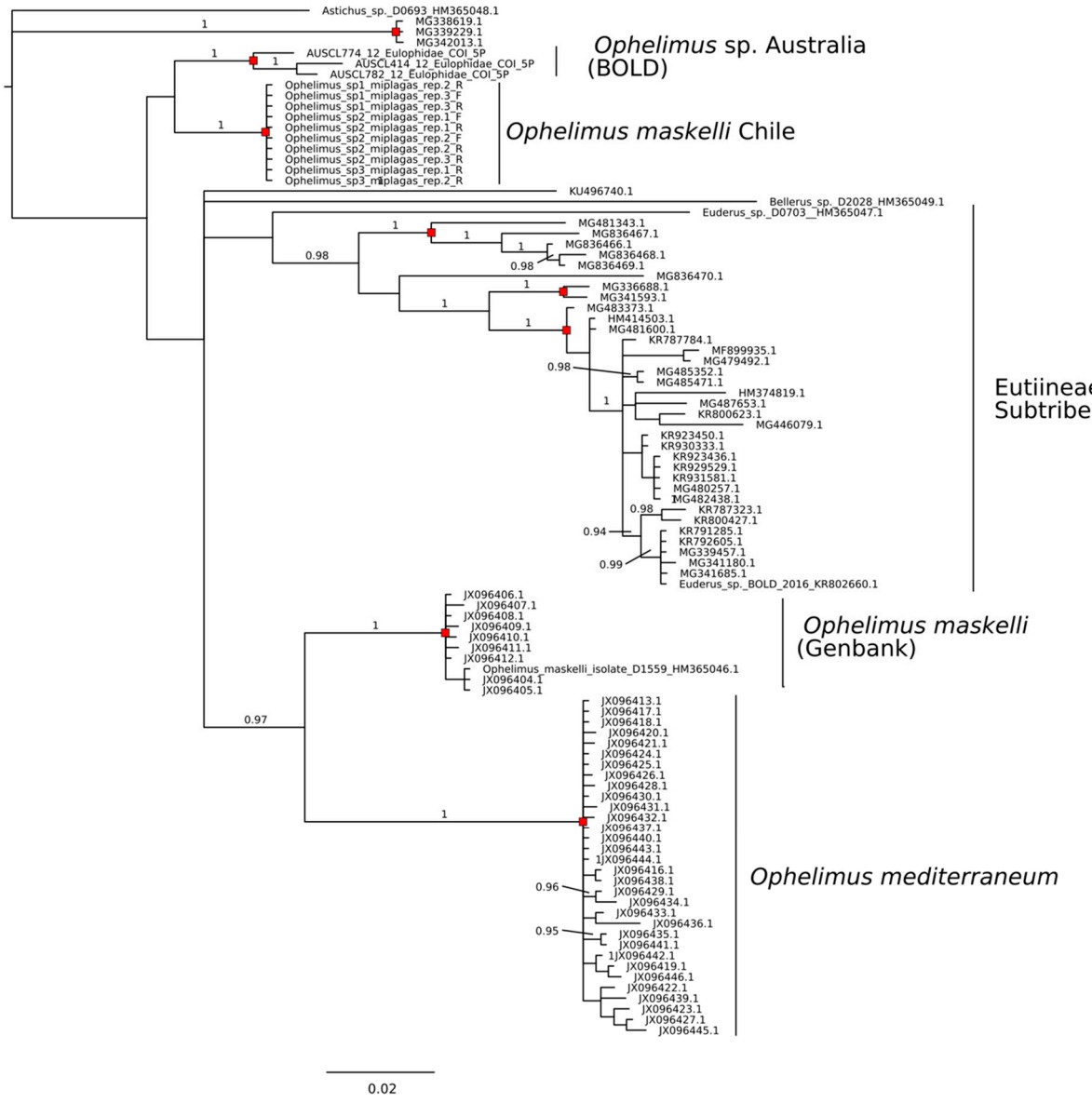

**Figure 5.** Bayesian inference (BI) consensus tree from Mr. Bayes. Red squares represent proposed groups made by mPTP. Significant posterior probability values (*PP* > 0.95) are shown above branches.

*3.4. Sequence Delimitation COI*

The analysis carried out on the trees obtained with ML and BI criteria revealed very similar delimitation proposals. In the case of the Chilean sequences, they formed a discriminable group, together with the Australian sequences in the ML analysis (Figure 4). In contrast, analyses with BI suggested both groups to be different entities (Figure 5). In the case of the rest of the registered sequences, *Ophelimus maskelli* was detected to be different from *O. mediterraneus* (Figures 4 and 5). For the rest of the groups, associations were differentiated for *Astichus* sp. and within Eutiineae; although, the number of detected groups depended on the analysis criteria used (ML or BI, Figures 4 and 5).

In the case of analysis of exclusive *Ophelimus* sequences, four differentiable groups were detected, which received different levels of grouping support and delimitation depending on the choice of molecular clock used. In the case of strict clock, a very significant grouping was detected between the Chilean and Australian sequences of *Ophelimus* (*PP* > 0.99, Figure 6), which also showed high levels of significance for delimitation in the bGMYC (*PP* > 0.99, Figure 6). In the same analysis, the recorded sequences of *O. maskelli* and *O. mediterraneus* were not recovered as monophyletic, and each group

demonstrated a high level of individual significance in their discrimination ($PP > 0.99$, Figure 6). For the lognormal flexible clock, the grouping of Chilean and Australian sequences was maintained with high levels of significance ($PP > 0.99$, Figure 7), although only the Chilean sequences showed a distinguishable level of discrimination support ($0.9 > PP > 0.95$, Figure 7). In the case of the other sequences, a group that is not significantly supported is distinguished ($PP < 0.95$, Figure 7), and it is individually possible to discriminate entities, but with a low level of intergroup delimitation ($0.5 > PP > 0.9$, Figure 7).

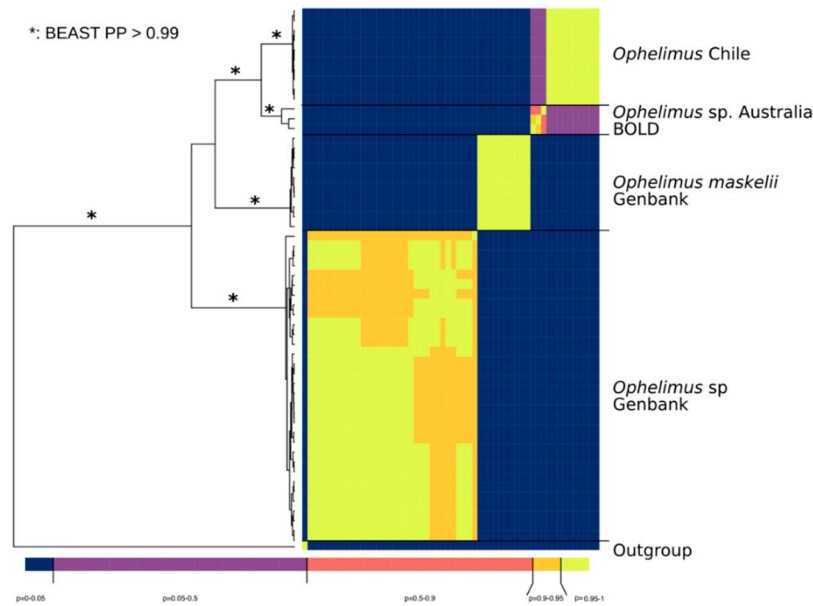

**Figure 6.** Ultrametric tree of inferred genealogical relationships with *COI*. For this tree a strict molecular clock was used in BEAST. Only those groupings with a very high significance are highlighted ($PP > 0.99$).

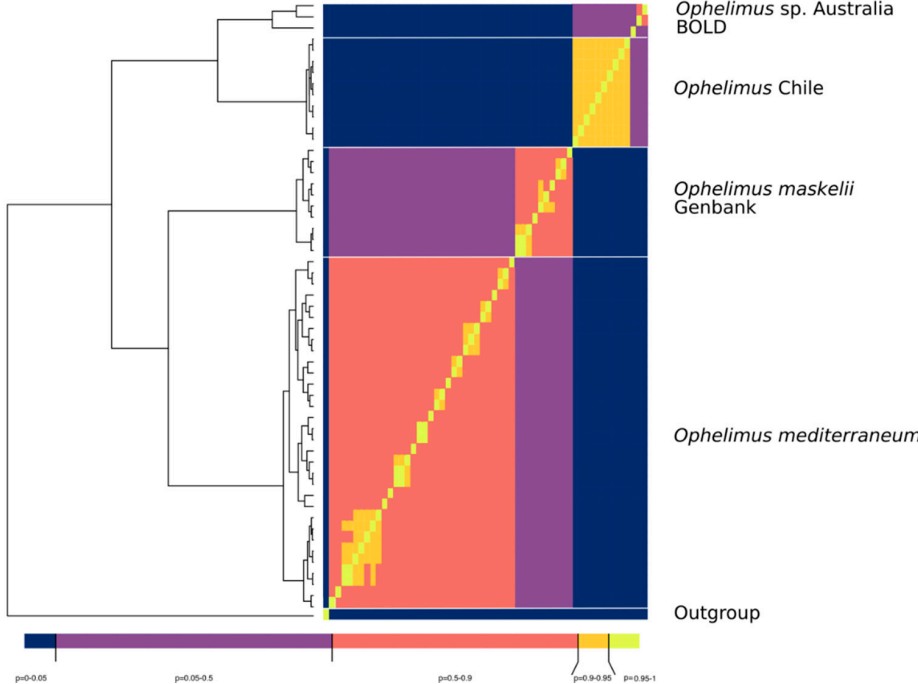

**Figure 7.** Ultrametric tree of inferred genealogical relationships with *COI*. For this tree a lognormal flexible clock was used in BEAST. Only those groupings with a very high significance are highlighted ($PP > 0.99$).

## 4. Discussion

*Ophelimus migdanorum* nov. sp. induces galls on stems, petioles, laminae and leaf venations of *E. globulus* and *E. camaldulensis* [35]. Its origin is unknown; it is assumed, however, that it is a species from Australia, since it is an insect that attacks *Eucalyptus* species [7,8] (La Salle, personal communication). The morphological description of this species, based on individuals taken from galls developed on *E. globulus* in the six sampling areas (between the Valparaíso and Los Lagos regions), allowed us to allocate it taxonomically within the genus *Ophelimus* and distinguish it from other congeners, confirming its morphological status of undescribed species.

*O. migdanorum* nov. sp. has three rings in the antenna, while *O. maskelli* has four, according to description by Protasov et al. [6] (Table 3). The rings of the antenna would be a distinctive feature of Eulophidae, being able to vary their number between two and four, and exceptionally, five [36].

Until the present study, the fundamental morphological characteristic that had enabled the differentiation of the agallicolous species of the genus *Ophelimus*, is the presence of SSV(s), since the difference in size and color are almost imperceptible [8]. In this study, smaller individuals (females 0.712–1.105 ± 0.10087 mm and males 0.781–1.144 ± 0.10820 mm long) were observed having an SSV; another larger group (females 0.846–1.253 ± 0.11845 mm and males 0.945–1.346 ± 0.12806 mm long) had two SSVs; and finally, another group of the largest individuals (females 0.930–1.413 ± 0.11196 mm and males 1.118–1.841 mm long) had three SSVs. Based on that and considering that *O. maskelli* presents only one SSV [6], it was initially estimated that the first group could correspond to *O. maskelli*, not reported in Chile, and the second and third groups to another undescribed agallicolous *Ophelimus* species. Discarding, in all cases *O. eucalypti*, for their presenting four or more SSVs [8]. However, our subsequent descriptive results would indicate that, consistently, the presence of SSVs would not be a morphological feature that could distinguish between *O. maskelli* and *O. migdanorum*, and that the antenna morphology would complement the distinction between both species, even among individuals presenting a single SSV. This situation confirms the scarce and incomplete information available for the species of the genus *Ophelimus*. What exists is based on the unfinished manuscript of Girault, a description of *O. maskelli* [6], and the limited available taxonomic information of *O. eucalypti* [56].

The result of the analysis of the molecular sequencing was mostly consistent with the morphological results, showing that the three types of specimens detected in Chile (with one, two and three SSVs) corresponded to the same phylogenetic unit, or that they differed from *O maskelli* and other *Ophelimus* sp described in the genetic data bank. Also, the recovery of a monophyletic group with high support between approximate delimitation would confirm that the *COI* sequences belonging to Chile would correspond to a new species, operationally differentiable in a phylogenetic context. Interestingly, the results were less clear when trying to suggest the monophyly of *Ophelimus* (Figures 6 and 7). This result could be explained by the lack of phylogenetic information observable from the *COI* marker, which requires the inclusion of additional regions to improve its resolution at genus and tribe scales [10]. In spite of this result, it is very probable that the *COI* region will serve as an effective vessel for the search and identification of new entities in *Ophelimus* (known particularly for its notorious resolution at interspecific scales) and other related genera.

The determination of *O. migdanorum* nov. sp in the six sampling locations, accounts for its dispersion and establishment in the country, between the regions of Valparaíso in the north and the Los Lagos in the south [35]. The wide dispersion could be explained by the long time elapsed, 14 years, since its detection as *Ophelimus*. sp [35], and by the continuity of the plantations of *E. globulus* present in the country that reach an estimated area of 563,000 ha in the study area, equivalent to 95% of the country surface covered by this species [57]. This biotic agent represents a new threat to the sustainability of *E. globulus*'s cultivation in the country, adding to the effects of drought and to the damage caused by *Gonipterus platensis*—Detected in Chile in 1998 [58].

The *Ophelimus* species are all associated with eucalyptus galls and, in general, they are all considered to be gall-inducers [12], affecting more than 90% of plantations in Chile [59]. The level of damage was not evaluated in this study; however, during the sampling process, gall formation

was observed in succulent stems, petioles and in the leaf rib, and less frequently, in the leaf. Those formations are associated with the mortality of leaves and branches, mainly in the middle third of the height of the trees, but even with the mortality of the trees themselves (consistent with the observations made by Bain [37] in New Zealand for the damages caused by *O. eucalypti* on *E. globulus*).

The results constitute a concrete contribution to the knowledge of the *Ophelimus* genus, both in morphological, molecular and behavior aspects. In the case of morphology only *O. maskelli* and *O. mediterraneus* had been described with publicly available information. In the Genbank, there were only 91 sequences of mitochondrial gene *COI* for the genus *Ophelimus*, and ten were added. Finally, BOLDSYSTEMS recorded only seven records of Ophelimus sp. with public sequences, forming only one BIN (cluster), but this study increased the number to eight records and two BINs.

## 5. Conclusions

Specimens of *Ophelimus* detected in Chile constituted a new species or taxa that had not previously been described. It should be noted that the present study was limited only to the discovery of a new polyphenic entity in *Ophelimus*, but that its systematic and taxonomic context, of this and other entities, is far from being completely clarified. An important element is the lack of information about the taxonomic integrity of other *Ophelimus* species, which mostly do not have comprehensive systematic studies. Considering the importance of this and other genera as invasive species with high economic impact for forestry and agricultural activities, it is necessary to encourage a better and more taxonomic treatment of these groups. The present work based on molecular markers proves that it is possible to distinguish and discover entities using complementary characteristics, and support those generated by traditional morphology. Therefore, it is expected that this work will serve as an example of the need to continue deepening the entomological study of *Ophelimus* and the improvement in the development of the systematization of the groups studied.

**Supplementary Materials:** The following are available online at http://www.mdpi.com/1999-4907/10/9/720/s1, Table S1: Characters used for the morphological description of Ophelimus detected in Chile, according to Burks et al. (2011); Table S2: Length of the sub marginal vein of *Ophelimus migdanorum* nov sp: 1, 2, and 3 SVSs. Table S3: Dimensions of morphological structures for individuals of *Ophelimus migdanurum* nov sp, for 1, 2 and 3 setaes. Tabla S4: Provision of the Holotype, alotype and paratypes, of *Ophelimus migdanorum* nov sp. Table S5: Analyzed *COI* sequences. Figure S1: Location of *Ophelimus nov sp*, Chile, in dendrogram made from the characteristics according to Burks 2011. Figure S2: Length of insects, males and females, with 1, 2 and 3 mushrooms in the SVSs.

**Author Contributions:** Conceptualization, G.M.-M., A.O.A., R.H., O.T.-N., H.A.B., and E.R.; data curation, G.M.-M., R.H., and O.T.-N.; formal analysis, G.M.-M., R.H., and O.T.-N.; research, G.M.-M., R.H., and O.T.-N.; methodology, G.M.-M., R.H., and M.C.-S.; visualization, G.M.-M.; writing—original draft, G.M.-M., T.S.O., M.C.-S., R.H., and O.T.-N.; writing—review and editing, G.M.-M., T.S.O., M.C.-S., R.H., O.T.-N., H.A.B., E.S., and E.R.

**Funding:** This work was financed by Integrated Management of Plagas Ltda. (*MIP*lagas Ltda.). The funder provided support in the form of salary, sample collection, use of company premises, use of equipment for identification, photographic and measurement material, data collection, parenting from the author G.M.-M., but did not have any additional role in the design of the study and the analysis, the decision to publish, or the preparation of the manuscript. The specific functions of these authors are articulated in the section "Author Contributions".

**Acknowledgments:** To John La Salle (in memoriam) for the first identification of the biological material and the motivation to identify the new *Ophelimus*. To the *MIP*lagas team, for their support, concern, collection of material, and measurements. To Paula Borrajo for the gall photos of *Ophelimus maskelli*. To Felipe Vargas for statistical analysis. To Miguel Castillo and Angela Sierra-Almeida, for general support and correction of this writing. To the Advanced Microscopy Center CMA Biobío of Universidad de Concepción, for the photographs in the scanning electron microscope, and its willingness to cooperate regarding the photographs for this work.

**Conflicts of Interest:** The authors declare no conflict of interest. This work was financed and supported by *MIP*lagas Ltda. Gloria Molina-Mercader owns the company. There are no patents, products in development, or products being marketed to be declared. The finance and support does not alter our adherence to all FOREST policies on the exchange of data and materials.

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
