# Peer review of "Ophelimus migdanorum Molina-Mercader sp. nov. (Hymenoptera: Eulophidae): Application of Integrative Taxonomy for Disentangling a Polyphenism Case in Eucalyptus globulus Labill Forest in Chile"

_forests, doi:10.3390/f10090720_

Round 1
Reviewer 1 Report
The manuscript submitted by Molina-Mercader et al. is, fundamentally, a description of a new species of Ophelimus that appears to have established in Chile. The authors, probably correctly, presume that the species originated in Australia and has invaded Chile. It may or may not be present in other parts of the world and the results presented in the text provide some taxonomic details to distinguish this new species from the more widely distributed congener. The methods for establishing the identity of the species are the currently accepted standard combination of morphological and molecular approaches. That is, there does not appear to be anything particularly novel in the approach, despite it being highlighted in the title. They do suggest that they conducted some type of breeding tests (e.g., Section 2.3 Laboratory Breeding), but in fact, this is misleading use of language; the breeding chambers are simply emergence chambers to collect wasps emerging from field collected branches. The biggest problem with the manuscript is the English use (e.g., l. 342 and 343 use of the word "gentleness" when the conventional term for acknowledging a photograph source would be "courtesy of". The manuscript requires extensive editing for language and scientific convention. For example, in the Introduction, Eulophidae is italicised sometimes (by convention, taxonomic Family names are not italicised) and not at others. The same issue exists for some subfamily names. The acronym for the Commonwealth Scientific and Industrial Research Organization is CSIRO and not SCIRO (l. 117).
Author Response
First of all, we would like to thank the insightful comments and observations provided by reviewer 1, which concerns methodological observations provided in the present manuscript.
Related to the first comment pointed by the reviewer (“That is, there does not appear to be anything particularly novel in the approach, despite it being highlighted in the title”), the novelty is the use of integrative taxonomy to solve a case of species outside its natural habitat. None of the three aspects alone (morphological, phylogenetic and behavior) would have been clear enough to affirm that we are facing a new species.
Related to the second comment pointed by the reviewer (“They do suggest that they conducted some type of breeding tests (e.g., Section 2.3 Laboratory Breeding), but in fact, this is misleading use of language; the breeding chambers are simply emergence chambers to collect wasps emerging from field collected branches”) was replaced in all the manuscript “breeding chambers” by “emergence chambers”.
Related to the third comment pointed by the reviewer (“The manuscript requires extensive editing for language and scientific convention.”) we work intensely on improving writing, English and scientific convention. I hope to meet with the standards of this prestigious journal.

Reviewer 2 Report
The authors have demonstrated that the species they found infesting Eucalyptus in Chile is different from Ophelimus maskelli and probabyl represents a new species (although to be sure about this would require comparison with the other described Ophelimus species).
It is not clear why the phylogenetic analyses were conducted because they are not relevant for the taxonomic treatment, and given the poor taxonomic coverage of the family Eulophidae with few sequences that were analysed are not sufficient to provide any insights into eulophid phylogeny. The authors should rather provide more information about the DNA barcode gap and how this relates to the morphological differences between species (i.e. if DNA barcoding supports the morphological species delimitation or not). For taxonomic purposes a neighbor-joining tree based K2P distances is more appropriate than phylogenetic analyses using ML and BI (which would be more suitable for a phylogenetic study).
The authors state that the sequences were uploaded to GenBank and BOLD but no GenBank accession numbers or BOLD identifiers are given (Table S4 does not provide this information for the new species).
It seems that one sequence (Ophel_001) of the new species is present in BOLD. A quick search in BOLD revealed that there are other specimens with the same sequence (and therefore probably of the same species) from Bogota), so it seems that the species occurs also in Colombia (but this is not mentioned in the manuscript).
Despite this and several other shortcomings of the manuscript it should be published after the authors have addressed the reviewers comments properly.
Also the authors should correct the many misspellings (I found several in the abstract alone, e.g. CSIRO (not SCIRO), Ophelimus not italics, behavior (not behauvior) and even in the titel (sp. nov. not sp. Nov.)
Author Response
First of all, we would like to thank the insightful comments and observations provided by reviewer 2, which concerns methodological observations provided in the present manuscript.
Related to the first comment pointed by the reviewer (“the delimitation of Ophelimus maskelli should be corroborated using a larger number of species of Ophelimus”), we agree with this statement. Actually, one of the main difficulties to conduct this study was the sparse and incomplete sampling of COI sequences available for the genus and the tribe Eutiinneae. However, while some criticism can be cast on the fact that limits to O. maskelli cannot be properly detected without a proper sampling, our results are highly clear supporting that this taxon should not be labeled as O. migdanorum. Direct comparison among sequences (plus evidence of their separate occurrence across continents and clear differences in morphological assessment) highlights this point clearly, providing compelling evidence to, at least, separate samples of O. migdanorum from O. maskelli. We do not contradict questions of the limits of this or the rest of the Ophelimus taxa, which evidently requires a thorough systematic analysis to clearly define a robust taxonomic proposal. Instead, we agree that much more work is still needed at different levels to reach a clear consensus about the integrity of the current systematic proposal of the group.
Related to the second point (Why to elaborate a phylogenetic tree using such a wide and incomplete sampling of COI sequences from Eutinneae), this statement can be rubuted using the arguments provided in the first point. One of the first issues found at the time to elaborate this work is that, while morphological evidence is clear predicting putative groups, molecular evidence is still highly inconclusive to support Ophelimus as a monophyletic group. Previous studies (Bourke et al. 2010), using a multilocus approach, revealed very weak support this statement. As such, we defined a two-fold approach to clarify (and successfully revealed) that O. migdanorum is actually a delimited entity, at both generic and specific levels. Hence, we consider more robust and complete to consider evidence of integrity in a wider taxonomic scope. We do not aim to reconstruct Eulophiide phylogeny and we have not stated this objective in our manuscript.
On the concerns on the treatment of data (“The authors should rather provide more information about the DNA barcode gap and how this relates to the morphological differences between species”), we agree in this point and changes have been produced in the manuscript to clarify this point. Nevertheless, we consider that evidence already provided in the manuscript is enough to elaborate and corroborate this point. In the same line, we do not agree with critisism on the methodology employed to delimit entities (“For taxonomic purposes a neighbor-joining tree based K2P distances is more appropriate than phylogenetic analyses using ML and BI”) because two reasons. First, given the complexity of Eulophiide systematics and the lack of evidence of Ophelimus as genus, we see necessary to include a wider taxonomic sampling to corroborate both integrity and identity of O. migdanorum. For this case, the use of a phylogenetic framework becomes a more appropriate approach to control aspects of homoplasy, apparently widely present in our study group (Bourke et al. 2010). Second, our study uses a phylogenetic perspective of species, which differs from similarity-based methods in the direct corroboration of ancestor and descendent relationships. We consider this approach more robust to in the identification and delimitation of entities, as it allows to incorporate (or at least detect) effects associated to incongruity in signals of divergence not compatible with similarity based methods. This is evident in the use of coalescent based methods (GMYC and mPTP), which necessarily require a good estimation of branch method and it is not necessarily accommodated with more simple substitution models like K2P.
In the fourth point (“The authors state that the sequences were uploaded to GenBank and BOLD but no GenBank accession numbers or BOLD identifiers are given”), we did not consider to report identity of sequences necessary until obtaining confirmation of approval of our manuscript. We do apologize in case that we created a misunderstanding stating that the sequences were already uploaded to available databases. Yet, we follow a procedure usually standard for this type of publication as seen in more systematic or evolutionary journals. During these last days, the sequences were sent to the NCBI and are in process (Submission # 2254296).
Related to the identity of sequence Ophel_001, we were aware of its presence in BOLDsystem and its origin from Bogota. Nonetheless, we were not aware of the details as access to this entry was private when our query was conducted at this data set. Considering the evidence exposed, we believe relevant to reveal this information related to the presence of Ophelimus in SouthAmerica; yet, this does not compromise the integrity and sense of our results on O. migdanorum in Chile.
